# Dimethyl Fumarate Promotes the Survival of Retinal Ganglion Cells after Optic Nerve Injury, Possibly through the Nrf2/HO-1 Pathway

**DOI:** 10.3390/ijms22010297

**Published:** 2020-12-30

**Authors:** Sotaro Mori, Takuji Kurimoto, Hidetaka Maeda, Makoto Nakamura

**Affiliations:** 1Division of Ophthalmology, Department of Surgery, Kobe University Graduate School of Medicine, 7-5-2 Kusunoki-cho, Chuo-ku, Kobe 650-0017, Japan; smori@med.kobe-u.ac.jp (S.M.); manakamu@med.kobe-u.ac.jp (M.N.); 2Maeda Eye Clinic, 1-1-1 Uchihonmachi, Chuo-ku, Osaka 540-0012, Japan; hmaeda@amber.plala.or.jp

**Keywords:** dimethyl fumarate, neuroprotection, optic nerve crush, retinal ganglion cells, NF-E2-related factor 2, heme oxygenase-1

## Abstract

This study aimed to verify whether dimethyl fumarate (DMF) promotes the survival of retinal ganglion cells (RGCs) after optic nerve crush (ONC) accompanied by activation of the NF-E2-related factor 2 (Nrf2)/heme oxygenase-1 (HO-1) pathway. We examined changes in the densities of tubulin β3 (TUBB3)-positive RGCs and the amplitudes of the positive scotopic threshold response (pSTR), reflecting the functional activity of RGCs, recorded on an electroretinogram, with daily administration of DMF, on day 7 after ONC. Furthermore, immunohistochemical and immunoblotting analyses were performed to study the activation of the Nrf2/HO-1 pathway using retinas treated with daily administration of DMF. Daily administration of DMF increasedthe density of TUBB3-positive RGCs in a dose-dependent fashion and significantly increased the amplitude of the pSTR. Immunohistochemical analysis showed that DMF administration increased the immunoreactivity for Nrf2 and HO-1, a potent antioxidant enzyme, in RGCs immunolabeled with RNA-binding protein with multiple splicing (RBPMS). Immunoblotting analysis revealed an increase in the nuclear expression of Nrf2 and marked upregulation of HO-1 after DMF administration. These results suggest that DMF has survival-promoting effects in RGC after ONC, possibly via the Nrf2/HO-1 pathway.

## 1. Introduction

Originally, dimethyl fumarate (DMF) and its derivative, monomethyl fumarate (MMF), were used as oral medications against psoriasis, chronic dermatitis, and multiple sclerosis; thus, their safety and effectiveness against these diseases have been established [1,2]. After engulfment, DMF is rapidly degraded to MMF, which is absorbed in the digestive tract. Previous studies using an animal model of experimental allergic encephalomyelitis (EAE) or cerebral vascular disturbance have shown that the pharmacological effects of DMF are mainly antioxidative and immune modulating.

The Kelch-like ECH-associated protein 1 (Keap1)/NF-E2-related factor 2 (Nrf2) pathway has been suggested to play a critical role in the antioxidant effects of DMF [3,4]. The cytosolic Nrf2 protein constantly binds to Keap1 under physiological conditions. However, under stress conditions, Nrf2 dissociates from Keap1 and is transferred to the nucleus. Subsequently, Nrf2 extensively upregulates a group of genes encoding enzymes, including heme oxygenase-1 (HO-1), nicotinamide nucleotide transhydrogenase oxygenase-1, and glutamate-cysteine ligase catalytic subunit, all of which play a pivotal role in the defense mechanism against mitochondrial dysfunction induced by oxidative stress or the reduced production of proteasomes.

DMF, producing another pharmaceutical effect, is a potent immune-modulating molecule mediated by the hydroxycarboxylic acid receptor 2 (Hcar2), which is a G-protein-coupled receptor (GPCR) [5,6]. In an ischemia-reperfusion brain model, DMF and MMF reduce the area of brain infarction and the degree of brain edema, whereby neuronal dysfunction secondarily improves; this process is mediated by the Nrf2/HO-1 pathway [7]. In an EAE model, DMF and MMF inhibit the recruitment of neutrophils into inflamed sites and improve the neurological deficit in wild-type mice. However, in Hcar2-knockout mice, the anti-inflammatory effects of DMF and MMF are significantly blocked [5]. In an experimental autoimmune neuritis (EAN) model, DMF modifies the polarity of invading macrophages from M1 to M2 and suppresses the production of inflammatory cytokines, including IFN-γ, IL-6, TNFα, and IL-17, resulting in axonal protection [8,9]. Furthermore, DMF can shift the immune response from Th1 to Th2 and then reduce the production of inflammatory cytokines [10,11].

Several studies using various animal models of a variety of ocular diseases have indicated that the Keap1/Nrf2 pathway is a critical signaling pathway against oxidative stress and inflammation [12,13]. Among them, studies of optic nerve injury have suggested that the Keap1/Nrf2 signaling pathway mediates the protective effects against retinal ganglion cell (RGC) death [14,15].

On the basis of the aforementioned pharmacological effects of DMF, we hypothesized that DMF has a beneficial effect on degenerated RGCs after optic nerve injury. Thus, in the present study, we examined whether DMF promotes the survival of RGCs after optic nerve crush (ONC) and activates the Nrf2/HO-1 pathway.

## 2. Results

Figure 1A shows fluorescent microphotographs of surviving anti-tubulin β3 (TUBB3) antibody-positive cells in flat-mounted retinas. Although retinas with ONC plus the vehicle exhibited a decrease in the number of surviving TUBB3-positive cells, DMF administration obviously increased TUBB3-positive cells (Figure 1A). Figure 1B shows the quantitative data of surviving TUBB3-positive cells in the presence of various concentrations of DMF. The mean densities of TUBB3-positive cells on day 7 after ONC significantly decreased compared with those of retinas with a sham operation (1598.7 ± 31.6 and 2924.3 ± 107.6, respectively; *n* = 6; *p* < 0.0001 repeated-measures analysis of variance (ANOVA)). In contrast, DMF administration significantly increased the mean densities of TUBB3-positive cells at 50 mg/kg (1942.2 ± 98.4) and 100 mg/kg (2176.8 ± 236.0) compared with those of retinas treated with the vehicle (both groups, *p* < 0.01; repeated-measures ANOVA). The survival-promoting effects were dose-dependent and peaked at a concentration of 100 mg/kg of DMF (Figure 1B).

Next, to verify whether DMF administration not only augmented RGC survival but also their functions after ONC, we examined the changes in the positive scotopic threshold response (pSTR), which reflects the activity of RGCs using the most effective concentration of DMF for promoting their survival. Figure 2 shows the representative waves of the STR at various strengths of light stimuli, as well as their quantitative data. At −5.2 log sc td s, there were no significant differences in the amplitudes of the pSTR between ONC with the vehicle and ONC with DMF (100 mg/kg; 5.2 ± 3.5 vs. 5.8 ± 3.1 µV; *n* = 6; *p* = 0.37, unpaired *t*-test). However, at −4.7 log sc td s, which is a brighter stimulation than −5.2 log sc td s, the amplitude of the pSTR in mice treated with DMF significantly increased compared to the vehicle (15.3 ± 3.6 vs. 11.0 ± 1.8 µV; *n* = 6; *p* < 0.05, repeated-measures ANOVA) (Figure 2A,C). A similar difference was reproduced at the brighter −4.2 log sc td s (38.2 ± 5.9 vs. 32.2 ± 3.0 µV; *n* = 6, *p* < 0.05, unpaired *t*-test) (Figure 2A,D).

Next, to examine whether DMF administration affects the expression of Nrf2 and HO-1 in RGCs after ONC, we performed immunohistochemistry (IHC) at the most effective concentration of DMF (100 mg/kg) for TUBB3 immunostaining analysis. Figure 3A,B shows the immunoreactivities of Nrf2 and HO-1, respectively, on day 7 after ONC in the presence or absence of DMF. DMF obviously increased the immunoreactivities of Nrf2 and HO-1 in the cells of the ganglion cell layer (GCL) compared with those of the sham operation and ONC with vehicle. Nrf2 and HO-1 immunoreactive cells were also positive for RBPMS, a marker of RGCs. Quantitative analysis revealed a significant increase in the immunofluorescent intensity in RGCs in the presence of DMF compared to the vehicle (Figure 3C).

Finally, to examine whether DMF upregulated the nuclear expression of Nrf2 and the retinal expression of HO-1, we performed Western blotting using cytoplasmic and nuclear fractions of retinal homogenates. Figure 4 shows the Western blotting data obtained on day 7 after DMF administration. DMF upregulated both cytoplasmic and nuclear expressions of Nrf2 (Figure 4A–D). Furthermore, DMF upregulated the crude retinal expression of HO-1 (Figure 4E,F).

## 3. Discussion

The present study demonstrated that DMF administration structurally and functionally promotes RGC survival after ONC in a dose-dependent manner. Moreover, we found that DMF administration increases the retinal expression of Nrf2 and HO-1, particularly in RGCs. These findings indicate that DMF exerts survival-promoting effects through the activation of the antioxidant pathway in RGCs.

The neuroprotective effects elicited by DMF on axon-injured RGCs are probably attributable to potent mechanisms, as follows: First, similar to the aforementioned evidence from studies that used cerebral ischemic or inflammation models, the antioxidative effects mediated by Keap1/Nrf2 after DMF administration are the most important mechanisms related to the neuroprotective effects in injured RGCs. Himori et al. revealed that the deletion of *Nrf2* enhances ONC-induced RGC death relative to that in wild-type mice. Treatment with 1-(2-cyano-3-,12-dioxolane-1,9 (11)-dien-28-oylimidazole (CDDO-lm), which is an Nrf2 activator, can significantly promote RGC survival in wild-type mice [14]. Furthermore, Fujita et al. demonstrated that transfection of *Nrf2* into selectively stressed RGCs promotes their survival after optic nerve injury [16]. However, in different types of retinal cells, Inoue et al. reported that treatment with RS9, which is an Nrf2 activator, protects retinal photoreceptor cells from the degeneration associated with light-induced retinopathy, accompanied by the upregulation of the nuclear expression of HO-1 [17]. Considering the results of the present study given this evidence, it is reasonable to consider that activation of the Keap1/Nrf2/HO-1 pathway by DMF is critical for the maintenance of the survival of not only RGCs but also photoreceptor cells. Secondly, activated microglia expressing Hcar2 have been suggested to be the key players in the neuroprotective effects induced by DMF. Jiang et al. revealed that MMF protects retinal photoreceptor cells from the degeneration associated with light-induced retinopathy. After light exposure, activated microglia expressing Hcar2 at a high level migrate into the retina, followed by the secretion of proinflammatory cytokines, such as TNFα, NF-κB, NLRP-3, IL-18, and IL-1β. MMF, an agonist of Hcar2, which inhibits the secretion of such cytokines, eventually results in the protection of retinal photoreceptor cells [18]. Similarly, robustly activated microglia are recruited into the retina after optic nerve injury [19,20]. Furthermore, several studies have revealed that the inhibition of microglial activation by minocycline provides protection against the secondary degeneration of RGCs after optic nerve injury [21,22,23]. Although we did not examine how DMF affects inflammatory responses in the retina after optic nerve injury, the neuroprotective effects elicited by DMF may include anti-inflammatory processes. Future studies are necessary to elucidate the relationship between the anti-inflammatory response and the neuroprotective effects induced by DMF administration in injured RGCs, as well as to further the exploration of the fundamental effects of DMF under physiological conditions. In addition, the efficacy of DMF administration in intractable optic neuropathies, such as glaucoma and Leber hereditary optic neuropathy, should be examined in the future.

## 4. Materials and Methods

### 4.1. Animals

This study was approved by the Animal Care Committee of the Kobe University Graduate School of Medicine and adhered to the ARVO statement for the use of animals in ophthalmic and vision research. Adult male C57BL/6J mice (aged 8 weeks) were purchased from CLEA Japan, Inc. (Tokyo, Japan). The mice were housed at the Kobe University Animal Facility with ad libitum access to food and water under a 12 h light/12 h dark cycle at room temperature (24 ± 2 °C). A total of 81 mice were used in the experiments. The present study was approved by the institutional review board of Kobe University Hospital (animal protocol number No. P170910-R1, 20 February 2018).

### 4.2. Drugs

DMF was dissolved in 0.8% methocel (Sigma-Aldrich, St. Louis, MO, USA) in phosphate-buffered saline (PBS) and applied at concentrations of 0, 25, 50, 100, and 150 mg/kg via oral gavage. PBS containing 0.8% methocel was used as the vehicle. DMF or the vehicle was first applied 30 min before ONC, and its application was continued between 9 and 11 a.m. for 6 consecutive days.

### 4.3. Optic Nerve Crush

The mice were anesthetized with an intraperitoneal injection of ketamine (100 mg/kg) and xylazine (10 mg/kg). Optic nerve surgery was performed, as reported previously [24,25]. Briefly, after resecting the superior rectus muscle, the left optic nerve was exposed and crushed with forceps 0.5 mm behind the eyeball for 10 s. In the sham operation, we only exposed the left optic nerve without crushing it.

### 4.4. RGC Labeling

To visualize RGCs, immunolabeling with an TUBB3 antibody (Biolegend, San Diego, CA, USA) was performed, as reported previously [26,27]. On day 7 after ONC, the mice were sacrificed and perfused using 4% paraformaldehyde (PFA). The retinas were dissected, post-fixed with 4% PFA, and washed with PBS. After blocking in PBS with 0.1% Triton X-100 (PBS-T) supplemented with 5% bovine serum albumin (BSA), the dissected retinas were incubated with an Alexa Fluor 488-conjugated anti-TUBB3 antibody (1:500) in PBS-T at 4 °C overnight. After washing with PBS, the retinas were flat-mounted on a glass slide, with the GCL facing up.

Images of TUBB3-positive cells at eight pre-specified flat-mounted retinal areas were acquired under a fluorescence microscope (BioZero, Keyence, Osaka, Japan), as follows: An observer (T.K.) blinded to the treatment conditions of the mice counted TUBB3-positive cells at two points (1 and 2 mm from the optic disc) in each retinal area quadrant (temporal, nasal, inferior, and superior) using a 40× objective lens, i.e., 0.145 mm^2^ per visual field and a total of eight visual fields per retina. On the basis of these images, averaged cell densities were calculated using Image J software (National Institutes of Health, Bethesda, MD, USA). We used six mice for each dose of DMF or the sham operation.

### 4.5. Electroretinography (ERG)

The mice were dark-adapted overnight before testing, and all procedures were performed under dim red light, as reported previously, with a modification [26,28]. Briefly, on the day of the experiment, the mice were anesthetized with an intraperitoneal injection of ketamine (100 mg/kg) and xylazine (10 mg/kg). A contact lens electrode embedded with gold wire was placed as an active electrode on the cornea (Mayo, Aichi, Japan), and a chloride silver plate was placed as a reference electrode in the mouth. A grounded aluminum sheet placed under the animal served as the ground electrode. The body temperature was kept at 37 °C with a heating pad. ERG was performed in both eyes simultaneously using a Ganzfeld bowl. Responses were amplified 10,000 times and band-pass-filtered from 0.3 to 500 Hz (PuREC PC-100, Mayo, Aichi, Japan). Moreover, responses were amplified differentially and band-pass-filtered at 0.125–50 Hz for STR recordings. Responses to flashes were averaged with an interstimulus interval ranging from 1 s for dim light to 10 s for the brightest flashes. To record STRs, serially increasing luminescence intensities of −6.2, −5.2, −4.7, and −4.2 log sc td s were used and the responses to 50 repeated stimuli for each intensity were averaged. We used six mice in each operation (sham operation and ONC with the application of the vehicle or 100 mg/kg DMF).

### 4.6. Immunohistochemistry

Retinal cryosections (8 µm thick) were collected on glass slides and fixed with 4% PFA for 10 min. After blocking with 5% BSA, the sections were incubated at 4 °C overnight with an appropriate concentration of primary antibodies, as listed in Table 1. After three 10 min washes with PBS-T, the sections were incubated with secondary antibodies at room temperature for 1 h. Thereafter, the sections were washed three times with PBS-T, counterstained with 4′,6-diamidino-2-phenylindole (DAPI; Vector Laboratories, Burlingame, CA, USA), and mounted with coverslips.

### 4.7. Immunofluorescence Analyses

The intensities of the fluorescence staining of RGCs and the inner plexiform layer (IPL) were evaluated, as reported previously [27]. As negative controls, we omitted primary antibodies. RGCs and the IPL were identified by RBPMS and DAPI staining. The mean immunofluorescence intensity in RGCs was calculated using ImageJ software based on 10 cells per section, 3 sections per case, and 3 mice per condition. The mean intensities of Nrf2 and HO-1 in the IPL were measured in 3 regions per case in 3 cases per condition using ImageJ software. Individual mean values were corrected by the staining levels in corresponding negative controls and then averaged across each group.

### 4.8. Western Blotting

Retinas were lysed in hypotonic lysis buffer containing 10 mM HEPES-KOH (pH 7.9), 10 mM KCl, 1.5 mM MgCl_2_, 1 mM dithiothreitol, 0.5 mM phenylmethylsulfonyl fluoride, and a protease inhibitor cocktail and then centrifuged at 10,000× *g* at 4 °C for 15 min. The supernatants were used as the cytoplasmic fraction. After removing the supernatants, the pellets including the nuclear fraction were incubated with a nuclear lysis buffer containing 20 mM HEPES-KOH (pH 7.9), 400 mM NaCl, 1.5 mM MgCl_2_, 0.2 mM EDTA, 1 mM dithiothreitol, 5% glycerol, and a protease inhibitor cocktail (Sigma-Aldrich) and further incubated for 30 min on ice. The lysates were centrifuged at 13,000× *g* at 4 °C for 5 min. The supernatants were used as nuclear fraction samples. Immunoblotting analysis of β-actin and TBP was performed to confirm the absence of contamination between cytoplasmic and nuclear fractions [29]. Retinal proteins (60 μg) were separated by SDS-PAGE (catalog #: XP08160BOX; Thermo Fisher Scientific, Waltham, MA, USA) and transferred to polyvinylidene difluoride (PVDF) membranes (catalog #:10600123; GE Healthcare Life Sciences, Buckinghamshire, UK). The membranes were blocked with 5% BSA in Tris-buffered saline containing 0.1% Tween 20 (TBST) at room temperature for 1 h, followed by incubation at 4 °C overnight with the primary antibody (against Nrf2, HO-1, β-actin, and TBP, as listed in Table 1) in TBST. After three 10 min washes with TBST, the membranes were incubated with horseradish-peroxidase-conjugated anti-rabbit IgG (1:2000) at room temperature for 1 h. After four 15 min washes, chemiluminescence detection was performed using electrochemiluminescence (ECL) reagents (catalog #: RPN2232; GE Healthcare Life Sciences), and the signals were quantified using β-actin and TBP expression levels as a reference. Signals were obtained and analyzed using the LAS-3000 Mini Digital Imaging System (FujiFilm, Tokyo, Japan). We used six mice for each operation (sham operation and ONC with application of the vehicle or 100 mg/kg DMF).

### 4.9. Statistical Analyses

Data are reported as means ± standard error of the mean. Statistical analyses were performed using Microsoft Excel version 2013 and MedCalc version 19.0.3 (MedCalc Software, Ostend, Belgium). Statistical comparisons were performed using one-way ANOVA with the Bonferroni’s test for post-hoc analysis when three or more groups were compared and with unpaired *t*-tests when two groups were compared. The significance level was set at *p* < 0.05.

## 5. Conclusions

In conclusion, systemic administration of DMF activates the Nrf2/HO-1 pathway in RGCs and promotes RGC survival after ONC. Considering the anti-inflammatory effects of DMF, this agent might have potent protective effects in a broad range of neurodegenerative diseases.

## Figures and Tables

**Figure 1 ijms-22-00297-f001:**
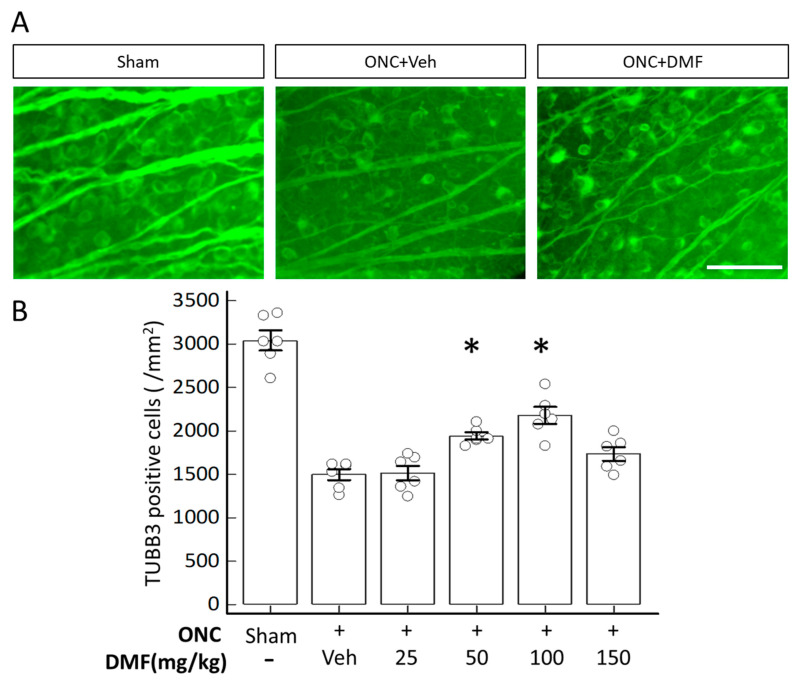
Dimethyl fumarate (DMF) administration promotes retinal ganglion cell (RGC) survival after optic nerve crush (ONC). (**A**) Photomicrographs of tubulin β3 (TUBB3)-positive cells with or without DMF administration on day 7 after ONC. The concentration of DMF was 100 mg/kg. The photomicrographs show temporal retinal areas at a 1 mm distance from the optic disc. (**B**) Quantitative analysis of surviving TUBB3-positive cells. The *y*-axis represents the mean density of TUBB3-positive cells. Sham indicates a sham operation without ONC. The asterisks indicate *p* < 0.01 (*n* = 6), and each circle denotes the values of individual animals. Statistical analysis was performed using one-way analysis of variance (ANOVA) with Bonferroni’s test for post-hoc analysis compared to ONC with the vehicle. Scale bar = 50 μm.

**Figure 2 ijms-22-00297-f002:**
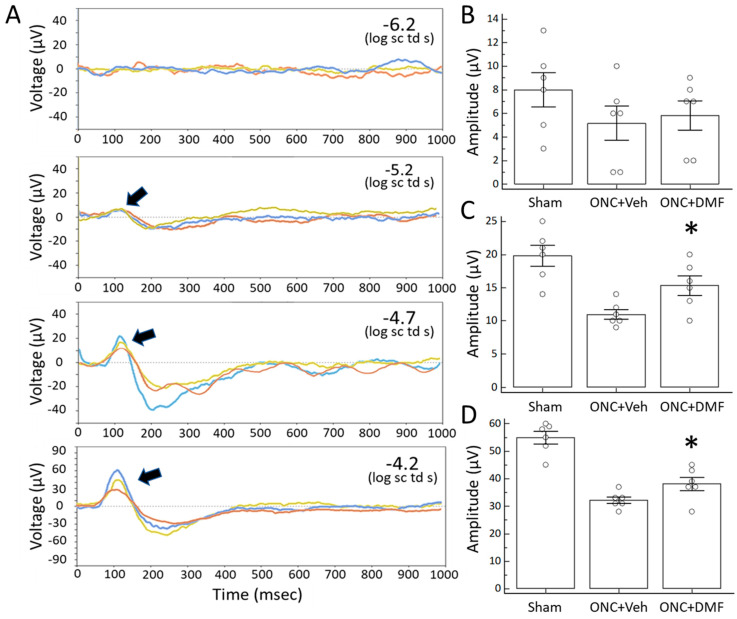
Scotopic threshold responses (STRs) elicited by a series of incremental dim-light stimuli. (**A**) Representative superimposed waves of STRs were recorded in mice (blue line, sham operation; red line, ONC with vehicle; yellow line, ONC with DMF at a concentration of 100 mg/kg). Arrows indicate the positive STR (pSTR). (**B**–**D**) Representation of the mean of the amplitudes of the pSTR in various dim-light stimuli. Each circle denotes the values of individual animals. (**B**) −5.2, (**C**) −4.7, and (**D**) −4.2 log sc td s. Asterisks indicate *p* < 0.05 (*n* = 6) compared to ONC with the vehicle (unpaired *t*-test).

**Figure 3 ijms-22-00297-f003:**
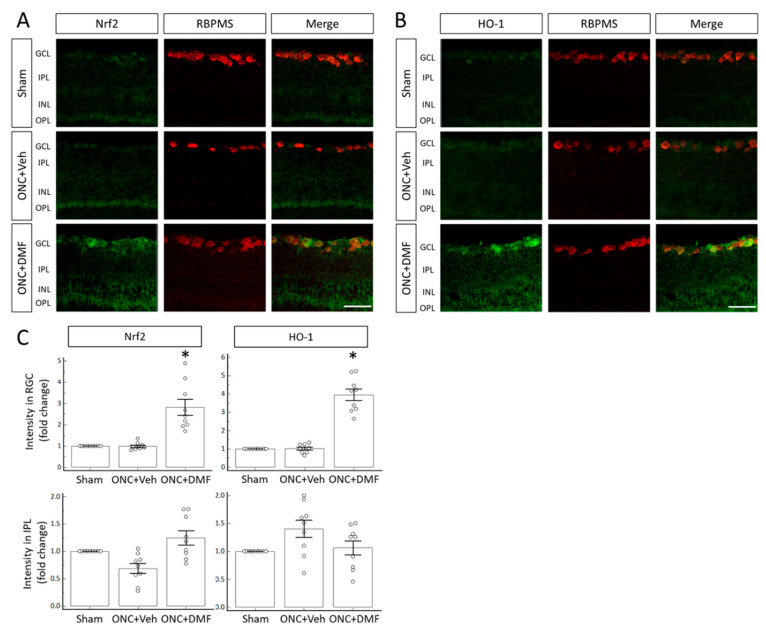
NF-E2-related factor 2 (Nrf2) and heme oxygenase-1 (HO-1) expression in the retina. (A) Immunoreactivities of Nrf2 and RNA-binding protein with multiple splicing (RBPMS). (**B**) Immunoreactivities of HO-1 and RBPMS. (**C**) Quantitative comparison of the immunoreactivities of Nrf2 and HO-1 in RGCs and the inner plexiform layer (IPL). DMF was administered orally at a concentration of 100 mg/kg. Each circle denotes the individual estimated values based on nine different retinal images (three different images from each animal). Asterisks indicate *p* < 0.05 (*n* = 9 each) compared to ONC with the vehicle (unpaired *t*-test). Scale bar = 50 μm.

**Figure 4 ijms-22-00297-f004:**
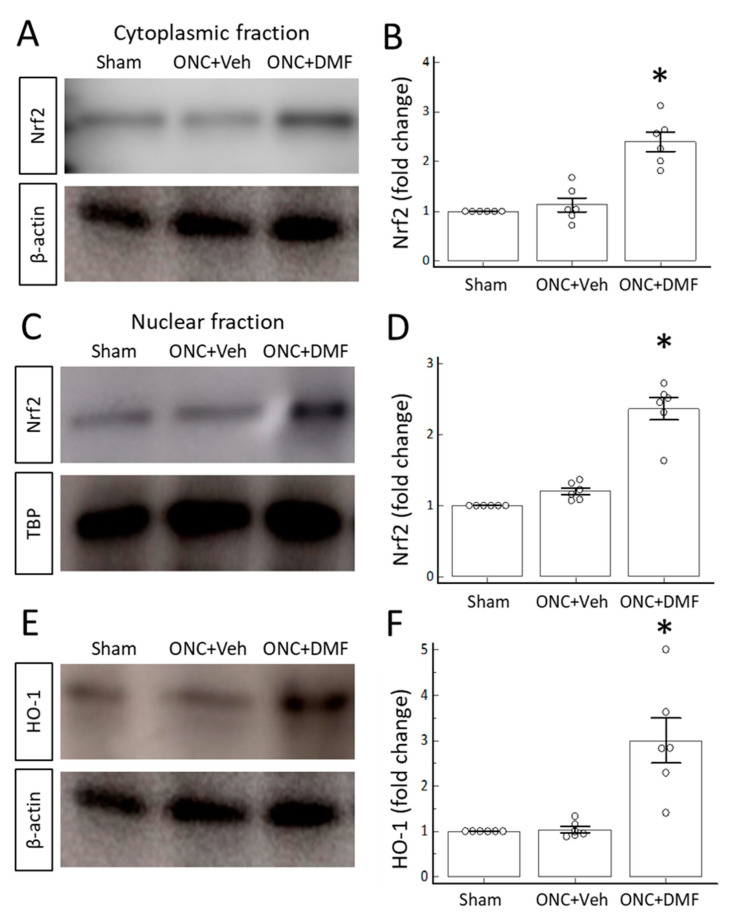
Western blotting analyses. (**A**,**B**) Nrf2 expression in cytoplasmic fractions extracted from retinal homogenates on day 7 after ONC with or without DMF: (**A**) representative blot; (B) quantitative comparison of the Nrf2/β-actin ratio normalized by that of the sham operation without ONC. (**C,D**) Nrf2 expression in nuclear fractions: (**C**) representative blot image. (**D**) quantitative comparison of the Nrf2/TBP ratio normalized by that of the sham operation without ONC. (**E,F**) HO-1 expression in the retina on day 7 after ONC with or without DMF: (**E**) representative blot; (**F**) quantitative comparison of the HO-1/β-actin ratio normalized by that of the sham operation without ONC. β-actin and TBP were used as internal controls. Each circle denotes the values of individual animals. Asterisks in all graphs indicate *p* < 0.05 (*n* = 6, unpaired *t*-test).

**Table 1 ijms-22-00297-t001:** List of antibodies used for immunostaining, immunoprecipitation, and Western blotting.

Antibody	Dilution and Purpose	Catalog No.	Company	Host
β-actin	1:200, WB	ab115777	Abcam	Rabbit
Nrf2	1:100 IP, 1:500 WB	ab137550	Abcam	Rabbit
HO-1	1:200 IP, 1:2000 WB	ab13243	Abcam	Rabbit
TBP	1:500, WB	GTX133204	Genetex	Rabbit
TUBB3	1:500, IHC	488–435L	Biolegend	Mouse
RBPMS	1:200, IHC	ABN1376	MerckMillipore	Guinea pig

TUBB3, tubulin β3; IHC, immunohistochemistry; IP, immunoprecipitation; WB, Western blotting; RBPMS, RNA-binding protein with multiple splicing.

## Data Availability

The data that support the findings of this study are available from the corresponding author, T.K., upon reasonable request.

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
