# Peer review of "Dimethyl Fumarate Promotes the Survival of Retinal Ganglion Cells after Optic Nerve Injury, Possibly through the Nrf2/HO-1 Pathway"

_ijms, 2020, doi:10.3390/ijms22010297_

Round 1

Reviewer 1 Report

The study investigates the effect of dimethyl fumarate (DMF) on the survival of retinal ganglion cells (RGC) after optic nerve crush. Mice were treated with DMF just before and after optic nerve crush. Immunohistochemistry was applied to study retinal ganglion cell density and changes in immunoreactivity using antibodies against proteins involved with the NRf2/HO-1 pathway. Electroretinography was applied to study RGC activity. Sham operated mice and mice receiving vehicle served as comparisons. The authors show that DMF treated mice have an increased RGC density compared to vehicle treated animals and an increase in the amplitude of the scotopic threshold response. The main finding is that DMF treatment increases the survival rate of retinal ganglion cells after nerve crush. The findings support the idea that DMF has a protective effect in neurodegenerative diseases.

The paper is clearly written and the results are well presented. It would be useful to have some more detailed information for the following points:

  1. According to the methods section, the results were obtained from retinal regions (dorsal, ventral, superior, inferior), but there is no reference to these different regions in the figures or the text. Please state whether or not there was a difference in different retinal regions.
  2. More importantly, please state in the legends which retinal regions are shown.
  3. Please define what “n=6, repeated measurements” means. Are these measurements from 6 different retinas?
  4. Please include the number of animals used in the method section.
  5. For clarity I suggest to delete the last sentence under section 4.3 and include the statement on perfusion with 4% PFA under section 4.4.
  6. The description of how the retinas were processed for immunohistochemistry is confusing. Line 19: “the retinas were then extended on a glass slide … and mounted”; line 193ff: “… the flat-mounted retinas were … mounted onto glass slides”

Author Response

According to the methods section, the results were obtained from retinal regions (dorsal, ventral, superior, inferior), but there is no reference to these different regions in the figures or the text. Please state whether or not there was a difference in different retinal regions.

More importantly, please state in the legends which retinal regions are shown.

Response: Thank you for the valuable suggestion. As indicated, we obtained the TUBB3 immuno-stained retina pictures at eight different points in each retina (described in lines 219–222). We apologize for not specifying the retinal area, wherein Figure 1A is represented in the legend. We have added the relevant sentence in the legend of Figure 1 (lines 84–85) in the revised manuscript. 

Please define what “n=6, repeated measurements” means. Are these measurements from 6 different retinas?

Response: We apologize for the unintentional ambiguity in our description. To clarify, “n = 6 repeated measurements’’ indicates that we obtained the data based on 6 different mouse retinas. Each circle indicates the value of individual mice. We have accordingly added a sentence in each figure legend (lines 87–88, lines 110–111, lines 127–129, and line 146) and the Method’s section (line 224, lines 239–240, and lines 286–287). 

Please include the number of animals used in the method section.

Response: Thank you for your suggestion. Reviewer 2 had raised the same concern as you. In response, we have now added the information regarding the number of animals we used in each of the figure legends (lines 87–88, lines 110–111, lines 127–129, and line 146) and in the Method’s section (line 224, lines 239–240, and lines 286–287). 

For clarity I suggest to delete the last sentence under section 4.3 and include the statement on perfusion with 4% PFA under section 4.4.

Response: Thank you for your suggestion. We have moved these sentences from Section 4.3 to Section 4.4 (lines 212–213) in the revised manuscript.

The description of how the retinas were processed for immunohistochemistry is confusing. Line 197: “the retinas were then extended on a glass slide … and mounted”; line 199ff: “… the flat-mounted retinas were … mounted onto glass slides”

Response: Thank you for your indication. We apologize for the ambiguity in the said sentence. We have rephrased these sentences to specify the process regarding immunohistochemistry in the revised manuscript (lines 214–217).

Reviewer 2 Report

The paper descrbes the protective effects of dimethyl fumarate in retinal ganglion cells. The paper is well written and contains the necessary information. The paper can be accepted after major revision. The biggest problem is with the number of animals, samples for which the accurate information is lacking from the manuscript.

In many places the authors refer to n=9 or 6 or 4, but the reader does not know wether these were all different animals, or the same, just for one experiment they used 9, for the other 6. Also, as these are retinal preparation, we do not know whether the shown results are from n=6 animals, or n=3 retinas, in which case the number of animals would only be 3 (2 retinas/ animal). This should be clarified in every figure legend and in the descriptuion of the methods for each experiment. How many animals were used altogether?

were the sham operated animals also treated with DMF or not? in case they were not, why? the effects of DMF should be clarified in normal retinas as well. As it is highly possible that DMF by itself increases the expression of those proteins, this should be tested.

Why did the authors use repeated ANOVA instead of one or two way ANOVA? (measurements were in different animals)

In Fig 2 legend unpaired t test is written, but repeated ANOVA in the text.

in Fig 1B, what are the circles in the bars? it should be indicated in the legend.

line 122: Figure 4 and F should be.

line 132: n=4 is indicated in the legend, yet there are 5 circles on the bars. Please clarify

line 182. sham should be included.

line 245: pellet should be instead of supernatant, as that contains the nuclear fraction.

Author Response

In many places the authors refer to n=9 or 6 or 4, but the reader does not know weather these were all different animals, or the same, just for one experiment they used 9, for the other 6. Also, as these are retinal preparation, we do not know whether the shown results are from n=6 animals, or n=3 retinas, in which case the number of animals would only be 3 (2 retinas/ animal). This should be clarified in every figure legend and in the descriptuion of the methods for each experiment. How many animals were used altogether?

Response: Thank you for your comment and suggestion. Reviewer 1 had also raised the same criticism as you. We apologize for our inconsistent and confusing description. We have rephrased these sentences to specify the number of animals used in each of the indicated figure legends and in the Method’s section (line 224, lines 239–240, and lines 286–287). In the last sentence of the legend to Figure 4, n = 4 was erroneously mistyped; the correct number is 6 (each circle indicates the values of individual animals). Again, we apologize for our overlook. We have now also added the number of mice used in each figure legends (lines 87–88, lines 110–111, lines 127–129, and line 146) and in the Method’s section (line 194) in the revised manuscript.

Were the sham operated animals also treated with DMF or not? in case they were not, why? the effects of DMF should be clarified in normal retinas as well. As it is highly possible that DMF by itself increases the expression of those proteins, this should be tested.

Response: Thank you so much for your insightful suggestion. We completely agree with your point that DMF probably increases the expression of Nrf2 and HO-1 proteins in the normal condition of the retina. In the present study, we intended to reveal the potent neuroprotective effects of DMF under pathological conditions such as axonal degeneration. In fact, Figures 3 and 4 illustrate that DMF significantly upregulated the expression levels of Nrf2 and HO-1 proteins relative to those of ONC with vehicle injection. Conversely, DMF showed no negative impacts on the expression levels of Nrf2 and HO-1 in RGC with optic nerve injury. Furthermore, several previous reports have revealed that the administration of DMF and its derivative, MMF showed no significant effects on the expression levels of Nrf-2 and HO-1 of intact mouse retina and brain (Casili et al., Journal of Neuroinflammation 17:59, 2020 in Figure 4A and C; Jiang et al., Invest Ophthalmol Vis Sci 60: 1275-1285, 2019 in Figure 6A). Thus, we considered it reasonable that the experiment setup to only examine the effect of DMF on the healthy retina was not a very essential requirement in the investigation of the effects of DMF under various pathological conditions. However, it is important to understand the physiological response of DMF on healthy cells or tissues. Based on this discussion, we have added the relevant information to the revised Discussion section (lines 183–184).  

Why did the authors use repeated ANOVA instead of one or two way ANOVA? (measurements were in different animals)

Response: We apologize for the ambiguity implied by our obscure sentences. We conducted one-way ANOVA, followed by Bonferroni’s test as a post-hoc analysis for multiple comparisons, as described in Section 4.9 of the revised manuscript. We have corrected these phrases in the revised Method’s section (lines 292–294).

In Fig 2 legend unpaired t test is written, but repeated ANOVA in the text.

Response: We apologize for the overlook. We conducted statistical analysis using unpaired t-test between ONC groups with vehicle and ONC with DMF. We have made the necessary corrections in the revised manuscript (lines 98, line 102).

in Fig 1B, what are the circles in the bars? it should be indicated in the legend.

Response: Thank you for your indication. Each circle denotes the values of individual animals. We have added this sentence to the legends of Figures 1 to 4 (lines 87–88, lines 110–111, lines 127–129, and line 146).

line 122: Figure 4 and F should be.

Response: We are not sure we understand the implication of this comment. We would like to respond to the comment if the reviewer elaborates the same for our better understanding.

line 132: n=4 is indicated in the legend, yet there are 5 circles on the bars. Please clarify

Response: Reviewer 1 has also indicated the same concern as yours. Accordingly, in the last sentence of the Figure 4 legend, n = 4 was mistyped and hence we have corrected the number to 6 in line 147 (each circle indicates the values of individual animals). We apologize for this mistake.

line 182. sham should be included.

Response: Thank you for your suggestion. I have added the relevant sentences about the sham operation in lines 207–208 of the revised manuscript.

line 245: pellet should be instead of supernatant, as that contains the nuclear fraction.

Response: Thank you for your suggestion. We have added a sentence regarding the collection of nuclear fraction in line 269 of the revised manuscript.

Reviewer 3 Report

This is a brief and concise study. The data presented here are of interest for specialist in the field. The experiments are designed well. Generating meaningful data by using different methods (ERG, Western blotting, immunohistochemistry) is sufficient.

It is a well-structured paper, and I suggest the manuscript to be accepted after the authors correct the mistakes in the manuscript.

Therefore, I suggest minor changes are required before the manuscript can be accepted. There are few mistakes in the manuscript; the authors should carefully read the revised version before resubmitting it.

Results section:

1. Fig1 B has a poor quality - Please correct the quality of this figure.

2. On the Fig1 B graph the dots for the reviewer are unclear, should explain a little bit more. Dots are showing the average of the whole mount retina/animal? Please clarify it.

3. line 84: Should discuss why the authors used "repeated measure ANOVA" statistics in this model. They used different animals in this experiment.Why the authors did not use ANOVA instead of repeated measue ANOVA?

4. line 84: Please clarify the N number. Is the N number  showing whole mount preparation or group? 

5. On the Fig2B,C,D: please change the 'SHAM ope' to 'SHAM'

6. lines 103 and 116: Please also clarify the N numbers.

7. line 122: 'and' is missing between A and D

8. line 132: N number is also not clear. The reviewer wonders if the N=4, why 5 dots were shown in the graphs (Fig4B,D,F)?  

9. Please discuss the groups used in the experment. Why the authors did not use SHAM+DMF-injected group? Does the DMF has any effects on the Nrf2 and HO-1 expression alone? It would be great to show the results of this group.

Methods section:

10. Please explain the number of animals used in all different methods

11. Please present the animal protocol number (ethical guidelines) in the Materials and Methods section.

12. line 245: Did the authors use the supernatans for the nuclear fraction samples, not the pellet? 

Author Response

  1. Fig1 B has a poor quality - Please correct the quality of this figure.

Response: We apologize for the blurred image. We have improved the quality of Figure1B images.

  1. On the Fig1 B graph the dots for the reviewer are unclear, should explain a little bit more. Dots are showing the average of the whole mount retina/animal? Please clarify it.

Response: Thank you for your suggestion. We have enlarged each dot. Each circle denotes the values of individual animals. We have added this sentence in the legend to Figure 1 (lines 87–88).

  1. line 84: Should discuss why the authors used "repeated measure ANOVA" statistics in this model. They used different animals in this experiment.Why the authors did not use ANOVA instead of repeated measue ANOVA?

Response: We apologize for confusing you with our obscure language. We performed one-way ANOVA, followed by the Bonferroni test as a post hoc analysis for multiple comparisons described in the last part of 4.9. We have corrected these phrases in the figure legend (lines 88–89) and in Methods section (lines 292–294).

  1. 4. line 84: Please clarify the N number. Is the N number showing whole mount preparation or group?

Response: Thank you for your query. Reviewer 2 also had raised the same question. We have added the sentence that we used six different mice in each procedure in the Methods section (line 224) and in the figure legend (lines 87­–88).

  1. On the Fig2B,C,D: please change the 'SHAM ope' to 'SHAM'

Response: Thank you for your suggestion. We have corrected this figure.

  1. 6. lines 103 and 116: Please also clarify the N numbers.

Response: Thank you for your suggestion. Reviewers 1 and 2 also had raised the same question. Each circle denotes the values of individual animals in Figure 2 and the individual estimated values based on nine different retinal images in Figure 3. We have added these comments to the figure legends (lines 110–111 and 127–129) and to the Methods section (lines 239-240 and 260).

  1. line 122: 'and' is missing between A and D

Response: We apologize for our mistake. This sentence indicated at Figure 4A, 4B, 4C, and 4D. We have rephrased this sentence (line 135).

  1. 8. line 132: N number is also not clear. The reviewer wonders if the N=4, why 5 dots were shown in the graphs (Fig4B,D,F)?

Response: Reviewer 1 has also indicated the same. In the last sentence of the Figure 4 legend, n = 4 is mistyped, and the correct number is 6 in line 147 (each circle indicates the values of individual animals). We apologize for our mistake.

  1. 9. Please discuss the groups used in the experment. Why the authors did not use SHAM+DMF-injected group? Does the DMF has any effects on the Nrf2 and HO-1 expression alone? It would be great to show the results of this group.

Response: Thank you so much for your insightful suggestion. Reviewer 3 had also indicated the same query. We completely agree with your point that DMF probably increases the expression of Nrf2 and HO-1 proteins in the normal condition of the retina. In this study, we aimed to reveal the potent neuroprotective effects of DMF under pathological conditions such as axonal degeneration. Actually, Figures 3 and 4 show that DMF significantly upregulated Nrf2 and HO-1 proteins expression levels compared with ONC with vehicle injection. Conversely, DMF had no negative impacts on the expression level of Nrf2 and HO-1 in RGC with optic nerve injury. Furthermore, several previous reports have revealed that the administration of DMF and its derivative, MMF, have no significant effects on the expression levels of Nrf-2 and HO-1 of intact mouse retina and brain (Casili et al., Journal of Neuroinflammation 17:59, 2020 in Figure 4A and C; Jiang D et al., Invest Ophthalmol Vis Sci 60: 1275-1285, 2019 in Figure 6A). Thus, we consider it reasonable that the experiment aimed only to examine the effect of DMF on the healthy retina is not such a high requirement in case of investigation for the effects of DMF under various pathological conditions. However, it is important to understand the physiological response of DMF on healthy cells or tissues. According to this discussion, we have added the same in the Discussion section (lines 183–184).  

Response:

Methods section:

  1. 10. Please explain the number of animals used in all different methods

Response: We have rephrased the sentence to specify the number of animals we used in each  procedure in the Methods section (line 224, lines 239–240, and lines 286–287).

  1. Please present the animal protocol number (ethical guidelines) in the Materials and Methods section.

Response: We have added this sentence in the Methods section (line 194–195).

  1. 12. line 245: Did the authors use the supernatans for the nuclear fraction samples, not the pellet?

Response: We apologized for confusing you with our obscure language. We have added a sentence regarding the collection of nuclear fraction in line 269.

Round 2

Reviewer 2 Report

the authors have improved the manuscript and have answered all questions and corrected the manuscript according to the suggestions. It can be accepted for publication